# Using Inertial Sensors to Determine Head Motion—A Review

**DOI:** 10.3390/jimaging7120265

**Published:** 2021-12-06

**Authors:** Severin Ionut-Cristian, Dobrea Dan-Marius

**Affiliations:** Faculty of Electronics, Telecommunication and Information Technology, “Gheorghe Asachi” Technical University, 679048 Iași, Romania; mdobrea@etti.tuiasi.ro

**Keywords:** head activity recognition, inertial sensors, wearable device, intelligent computing, tracking systems, metaverse systems, body motion recognition, motion detection, deep learning, machine learning, pattern recognition

## Abstract

Human activity recognition and classification are some of the most interesting research fields, especially due to the rising popularity of wearable devices, such as mobile phones and smartwatches, which are present in our daily lives. Determining head motion and activities through wearable devices has applications in different domains, such as medicine, entertainment, health monitoring, and sports training. In addition, understanding head motion is important for modern-day topics, such as metaverse systems, virtual reality, and touchless systems. The wearability and usability of head motion systems are more technologically advanced than those which use information from a sensor connected to other parts of the human body. The current paper presents an overview of the technical literature from the last decade on state-of-the-art head motion monitoring systems based on inertial sensors. This study provides an overview of the existing solutions used to monitor head motion using inertial sensors. The focus of this study was on determining the acquisition methods, prototype structures, preprocessing steps, computational methods, and techniques used to validate these systems. From a preliminary inspection of the technical literature, we observed that this was the first work which looks specifically at head motion systems based on inertial sensors and their techniques. The research was conducted using four internet databases—IEEE Xplore, Elsevier, MDPI, and Springer. According to this survey, most of the studies focused on analyzing general human activity, and less on a specific activity. In addition, this paper provides a thorough overview of the last decade of approaches and machine learning algorithms used to monitor head motion using inertial sensors. For each method, concept, and final solution, this study provides a comprehensive number of references which help prove the advantages and disadvantages of the inertial sensors used to read head motion. The results of this study help to contextualize emerging inertial sensor technology in relation to broader goals to help people suffering from partial or total paralysis of the body.

## 1. Introduction

Sensors are the most important component of intelligent devices; they can read and quantify information about the world around us. Sensor components have multiple applications, such as in home automation, elderly care applications, smart farming, automation, etc. Modern wearable devices are equipped with multiple sensors. To facilitate a sensor comparison in order to provide a comprehensive overview of sensing technology, the research community has tried categorizing the different types. Depending on different technologies, sizes, and costs, sensory technologies can be divided into three classes based on the sensed property (physical, chemical, or biological) [1]. This study focused on sensing technology used in head motion detection based on inertial sensors. In the last decade, head motion detection has been made possible through the use of various sensors, such as video-camera-based [2], radar-based [3], and radio-based [4]. To determine the best approach regarding multiple criteria, including cost, wearability, and classification performance, multiple surveys have been proposed in the literature. However, most studies have focused on determining the technologies and computational methods used for general human body motion [5].

Sensor placement [6], background clustering, and the inherent variability of how different people perform activities have been other areas of focus. Although the information from studies already conducted in this area is suggestive, a comparison between the applications of various detection methods to identify specific tasks performed by a certain part of the body with the help of inertial sensors is still missing. The current study focused on head motion technology based on inertial sensors. This idea can be considered a subdomain of the human activity recognition (HAR) technical field. This survey was proposed to provide helpful information for researchers and practitioners in the HAR domain who plan on using inertial sensors in head motion detection. This survey provides comprehensive information about solutions developed in the last decade. This information can be beneficial in many human-centric applications, such as home care support, gesture detection, and the detection of abnormal activities. The novelty of our proposal is that our study was focused on investigating the literature related to the existing methodologies applied to understanding the motion of a specific body part (in this case, the head). We have not yet discovered a similar approach in the literature, even for other human body parts. Most existing surveys focused on activity recognition methods, classification algorithms for activity recognition systems, or wearable inertial systems. To summarize the actual status of the literature, Table 1 presents 13 surveys relevant to understanding human motion based on inertial sensors.

This paper is organized as follows: Section 2 presents an overview of existing head motion detection applications based on inertial sensors; Section 3 provides a discussion on the literature review methods and review findings; Section 4 presents information related to different feature extraction and preprocessing methods; Section 5 presents a discussion on computational algorithms; and the discussion and conclusion are presented in Section 6 and Section 7, respectively.

## 2. Related Work

### 2.1. Head Motion Literature Overview

In the last decade, understanding human behavior has become an interesting research topic for researchers worldwide. Consequently, multiple methods have been proposed to detect and interpret body motion patterns. In this paper, our focus is to determine and provide an overview of the methods used in the literature for head motion detection based on inertial (IMU) sensors. An IMU typically has three types of sensors in its structure, or a combination of them, characterized by an accelerometer, gyroscope, and magnetometer sensor. The identification of human motion using IMU sensors has gained importance due to its small size and low manufacturing costs [18,19]. For this reason, the reading and classification of human motion patterns have become an exciting topic in the last decade. Most proposed studies focus more on the determination of general human activity and less on detecting specific activities (e.g., head motion, neck motion, foot motion, etc.). The position and number of inertial sensors used in motion recognition are essential aspects of human motion recognition systems from the perspectives of both cost and wearability.

In the literature, in the field of human activity recognition (HAR), the most commonly used sensor in the detection of human motion is an accelerometer. The performances reported based on this type of sensor are promising and have classification value up to 90% [8,9]. This observation is treated in most of the surveys proposed. Lara et al. [14] provide an overview of the features and computational methods used in motion recognition. Another study, proposed by Reich et al. [6], treats the problem of the number of inertial sensors included in HAR systems. This survey provides a good overview related to the position of the inertial sensor on the body, which is a common approach in HAR systems. Accordingly, it was observed that, in most cases, the inertial sensors are positioned on the chest, right wrist, right waist, or right upper leg. The least common approach was when the inertial sensor was positioned on a body extremity (neck, head, left ankle, etc.). This observation shows that the determination of a specific human activity in the HAR technical field is a new topic; few studies have addressed this issue. In addition, this leads to the fact that our proposal is unique and provides a good overview of the current status of head motion tracking systems based on inertial sensors. The topic of how many inertial sensors need to be used and positioned in motion tracking systems is also approached in the paper proposed by Bao and Intille [20]. Their study provides additional information about the accuracy of different existing solutions with a focus on the accelerometer sensor. Another topic approached by the existing surveys is related to the computational models used. Demrozi et al. [15] focused on the study of machine learning algorithms in developing HAR applications based on inertial sensors in conjunction with psychological and environmental sensors. According to their observation, the interest in deep learning (DL) has increased due to higher classification accuracy in the HAR systems. This positive result was obtained based on large activity datasets. In relation, most HAR results published in the last decade have focused on studying classical machine learning (CML) models. These computational models might be better suited due to the small training datasets, lower dimensionality of the input data, and availability of expert knowledge in developing the problem [21].

Another survey study, conducted by Nava and Melendez [16], analyzed the problem of the type of inertial sensor and system used in evaluating human activity. In their study, they suggested that it was possible to capture human activity through systems based on a single inertial sensor (accelerometer, gyroscope, or magnetometer), or on a complex unit sensing device based on the use of a sensor network. Most movement measurements are performed in the upper limb, lower limb, multiple limbs (upper and lower limbs at the same time), or in other body regions (head, trunk, back, or hip). According to their study, from 107 relevant papers, only 7 addressed the topic of monitoring body regions other than the upper and lower limbs, which is conventional. This fact highlights the contribution of this paper in the field of HAR systems. Previous observations related to human activity distribution systems are supported by a study published in 2020 by Rast and Labruyère [17]. Their survey focused on a literature review to determine the application of wearable inertial sensors used to monitor human motion. Consequently, from a total of 95 relevant papers, they concluded that an accelerometer and gyroscope represent the sensors used the most in detecting body motion.

Regarding sensor placement, the most common areas are the trunk and the pelvis. On the other hand, the most uncommon approach is to place wearable sensors on the head. This information highlights the contribution brought by this survey. This observation is also highlighted in Table 1, where 13 surveys relevant to human motion understanding based on inertial sensors are centralized. According to this information, most existing studies are focused on understanding general body motion, and very few on a specific part.

The following chapters will present the results obtained and the conclusion.

### 2.2. Taxonomy of Head Motion Analyses Using Inertial Sensors

Inertial sensors (i.e., accelerometers, gyroscopes, or magnetometers) have been used in the last decade in diverse applications in medicine or other interconnected fields. In Figure 1, the taxonomy based on existing applications in the field of head motion recognition is presented. The first class described corresponds to papers which are focused on proposing and designing proper techniques for measuring and quantifying head motion. The second case represents works based on the analysis and interpretation of classifying head motion activity. This step is possible by using a computational model (classical machine learning vs. deep learning). In addition, at this level diverse calibration computational methods beneficial for improving head motion classification performance are studied.

In the first class of our taxonomy work, we focused on designing head-wearable devices with diverse applicability. A common example is the detection of daily activities (e.g., sitting, standing, walking, etc.) [22,23], medical assistance (e.g., upper limb disabilities, elderly care, etc.) [24,25], or focusing on the understanding of head motion patterns [26,27]. For the second class, the published papers include an overview of the classification performance provided by computational models (classical machine learning vs. deep learning algorithms) [28,29], and various studies focusing on proposing and improving existing computational models [27,30]. Regarding the existing review studies focused on human motion understanding, most of them mainly address the device technology used [6], the placement of the IMU sensor [31], computational algorithms [32], inertial time series feature selection [32], or body rehabilitation [33]. All the previously mentioned categories are focused on general body inspection. In this study, our purpose was to review the technical literature to determine and present the existing state-of-the-art solutions in the field of head motion recognition systems. Thus, this survey focuses on determining the acquisition methods used, prototype structure, preprocessing steps, computational methods, and the techniques used to validate these systems. In addition, this paper provides a good overview of approaches, computational algorithms, and techniques used to monitor head motion with the help of inertial sensors in the last decade.

### 2.3. Literature Review Method

The literature review was performed on four relevant public databases: IEEE, Elsevier, Springer, and MDPI. Selection of the relevant papers was performed by applying the filter in the “Web of Science” core collection web page. To achieve this task, two groups of keywords were used in the structure of the final filter, i.e., Group 1 (“Head gesture” OR “Head” OR “Head HCI” OR “Head Motion Classification”) and Group 2 (“inertial sensor” OR “accelerometer” OR “gyroscope” OR “magnetometer” OR “IMU”). Consequently, a total of 1361 studies were reviewed on the basis of the criteria for inclusion/exclusion. For selecting the relevant papers for this survey, the publication date had to be within the period 2011 to 2021.

Applying the previously mentioned steps, we obtained 213 from the first selection. A full-paper revision was performed to finally select the most suitable papers for this study. Therefore, we selected only 51 papers. The stages of the paper selection process are presented in Figure 2. Most papers analyzed the head motion topic in the fields of electrical and electronic engineering, instruments and instrumentation, telecommunications, biomedical engineering, automation control systems, computer science and artificial intelligence, analytical chemistry, computer science information systems, applied physics, and robotics. The distribution of the relevant papers can be seen in Figure 3.

In Figure 3, the results presented are cumulated on the four public databases used (IEEE, Elsevier, Springer, and MDPI). Consequently, the general interest is to design and develop electrical and electronic systems for the automatization of several tasks of daily activities or to provide support for ill persons in a medical environment.

Since 2011, researchers’ interest in the field of head motion recognition has constantly increased. Excluding the current year, the trend in this area is expected to increase in the next years. This affirmation is supported by the graphic presented in Figure 4.

### 2.4. Review Findings

After the literature review, we noticed that the relevant selected papers had the following distribution: 49% focused on medical problems such as head tremor [34], cerebral palsy [30,31,32,33], fall detection [15,34], vestibular rehabilitation [35], physical and mental activity analysis [20,36,37], forward head posture [17,38], and musculoskeletal disorders [39]; 20% focused on the general problem of human–computer interaction [35,40,41,42]; and 10% focused on the development of new computational and calibration methods [39,43,44]. The last two topics were the development of sports devices (swimming, hokey, golf, motorcycle ride or spinning exercises: 10%) [45,46,47,48,49,50], and 8% focused on prevention and safety systems (drivers’ attention) [51,52,53]. All the selected papers used an electronic device with an inertial sensor placed on a specific part of the head. In this regard, the most common approach is the placing of the inertial sensor on the left or right forehead (20.73%), on the forehead (18.86%), and on top of the head (16.98%). Other commonly used areas are the back of the head (11.32%), the ear (11.32%), the eye or the neck (7.52%), and other areas on the head (5.64%). The anatomical distribution of inertial sensors on the head is illustrated in Figure 5.

Regarding the type of sensor, the distribution suggests that 31% of studies used a 9DOF(**D**egree **O**f **F**reedom) inertial sensor (accelerometer, gyroscope, and magnetometer), 25% used a 6DOF inertial sensor (accelerometer and gyroscope), and 21% used a 3DOF accelerometer inertial sensor, whereas the remaining studies used a 3DOF gyroscopic (4%), 3DOF magnetometer (4%), and inertial sensors working together with other types of sensors (8%), such as an EMG (electromyography) sensor [24], video camera [28], thermometer [36], or flex sensor [47].

## 3. Head Motion Systems

Head motion recognition systems rely on head activities which need to be recognized, and the type of sensor, its placement, and the complexity of head motion can affect the classification performance. For this reason, in the technical community, researchers meet several challenges in the construction of portable systems with low costs and adequate data acquisition, or in determining the proper signal attributes to be measured. Other challenges were related to computational performance improvements regarding the extraction of inertial features and designing the inference methods, and the recognition of a new user path without retraining computational models. The last challenges observed during the literature review were related to the implementation and evaluation of head motion systems in the online mode. In this section, we will present the existing approaches related to head recognition motion (HRM). The most important aspects of HRM systems are the motion sensors. Most proposed solutions are based on 9DOF inertial sensors (accelerometer, gyroscope, and magnetometer), 6DOF inertial sensors (accelerometer and gyroscope), and 3DOF accelerometer inertial sensors. Each type of inertial sensor provides several benefits in the process of head motion path detection. For example, an accelerometer sensor can measure the acceleration component; however, it cannot determine velocity or head positional changes with high precision. With gyroscopic sensors, the angular velocities can be recognized with high accuracy, whereas magnetometers can determine the head orientation with high accuracy, based on measurements of variation in the magnetic field values. The majority of proposed head motion devices focus on medical problems (49%). Cerebral palsy [38,54,55] is one of the most studied topics in HRM systems. Even in the last decade, multiple proposals have contributed to this area, with this topic remaining a challenging task for researchers. One study conducted by Rudigkeit et al. [38] investigated the possibility of controlling a robotic arm based on head motion. In their study, the inertial sensor included a 3D accelerometer, a 3D gyroscope, and a 3D magnetometer in its structure. Here, the sensor was placed on the head, aligned with the user’s spine. The acquisition and processing steps were performed on a desktop computer. In another study, Ruzaij et al. [29] proposed a method based on 9DOF inertial sensors to control a traditional wheelchair. In their study, they used an ARM (Acorn RISC Machine) microcontroller for the acquisition and computational steps. The inertial sensor, similar to the previous example [38], was placed on the head using a prototype headset. Another study analyzing the problem of cerebral palsy was conducted by Guilherme et al. [54]. In their study, the authors used an Arduino platform and MATLAB to simulate and test the real conditions of head motion. In addition to the 9DOF inertial sensors approach, we noticed that 6DOF inertial sensors (accelerometer and gyroscope) had been used to study cerebral palsy in the papers reviewed. Prannah et al. [37] proposed a prototype based on 6DOF IMU and Arduino platform. In their case, the sensor was placed on the head using a prototype headset. Evaluation of the compartmental behavior was performed using the Proteus program. In addition to cerebral palsy, in the field of head motion analysis there are several other topics which are beginning to be studied. These include head tremor [34], fall detection [40], sleep quality [56], general physical and mental activity analysis [41], and musculoskeletal disorders [39].

In a study conducted by Elble et al. [34], the authors analyzed the possibility of determining head motion tremor using a 6DOF inertial sensor placed on the vertex of the head. The tremor component in this case was detectable through calculations of the mean and maximum three-burst displacements in the spectral analysis. In determining body equilibrium, multiple methods have been proposed and studied based on the interpretation of head motion information. Such a system was proposed by Lin et al. [40] to determine involuntary fall disorder. Their method used inertial information provided by a 6DOF inertial sensor (accelerometer and gyroscope) with an additional magnetometer. The sensors were placed on the left ear using a self-designed eyeglasses prototype. The motion pattern was sent over a Wi-Fi connection using the cheapest ESP8064 module. In detecting falls, the system is capable of sending an alert to the emergency contact. The proposed system could observe the condition of portability for an estimated autonomy of at least 68 h. In their study, only 3 falls from 700 falling motions were selected. Another study related to fall detection was carried out by Chen et al. [25]. Similar to the aforementioned study, the inertial sensor was designed in a 9DOF topology and placed on the left ear. The inertial signals were sent to the computational block using a Wi-Fi module. Another study area is characterized by the identification of daily activities. Such a system was proposed by Cristiano et al. [23]. Their system contained a 6DOF inertial sensor and was designed to identify static and dynamic body activities. The inertial sensor was placed on the left side of the forehead. In a study published by Loh et al. [42], the authors studied the possibility of determining fitness activities. Here, the proposed device contained five inertial sensors. The inertial sensors were placed on a helmet, the left arm, left wrist, left pant, and left ankle. All inertial data were transmitted to a portable laptop. For validation, in addition to inertial sensors, they attached a video camera for tagging the activities. Regarding the sports activity, as well as the previous example, we observed that multiple solutions have been proposed to assist users in performing activities such as swimming [52], hokey [53], golf [57], or motorcycle VR [58]. Such a system was proposed by Michaels et al. [52] in order to coach new users to use inertial sensors. Their system used only a tri-axial accelerometer and considered the following requirements: to be lightweight, waterproof, have a long life battery, and to save acquired data for a long period. The HDM (Hardware Device Module) device was placed on the back of the athlete’s head, underneath the swimming cap. They indicated that this placement of the inertial sensor was relevant for their particular study, suggesting that the main disadvantage of their solution was related to the fact that fore–aft acceleration of the head was largely representative of the center of the body, whereas Euler angles (roll and pitch) are unrepresentative of the entire body. For data analysis and validation, the MATLAB development environment was utilized. Another category of applications related to head motion analysis is human–computer interaction (HCI) devices, which have general applicability [27,45,46,59,60,61,62]. The main purpose of most of these systems is to determine an accurate solution for the augmented reality field. Such a study was proposed by Young et al. [45]. They proposed and studied two interaction methods used to control a virtual object by combining touch interaction and head motions. The touch interaction was performed based on a nail-mounted inertial measurement unit. The head motion was tracked from inertial sensors built in an augmented reality (AR) environment on a head-mounted display (HMD), which could be used in a mobile environment. In another study conducted by Tobias et al. [46], the authors studied the topic of orientation estimation by combining information from a single inertial sensor located on the user head with inaccurate positional tracking. For their experiment, they used a smartphone with a 6DoF IMU sensor attached to an ARHMD device. To validate the experiment, they used an optical-laser-based system with an accuracy of <10 mm at 20 Hz.

Other categories related to head motion analyses which have been proposed in the last decade include safety devices used in the identification of drivers’ behavior with the purpose of avoiding car accidents. One such study was proposed by Han et al. [63], who studied the possibility of determining the driver’s posture. Detection of the driver’s postural path was possible through a tri-axial magnetometer attached on the back of the driver’s neck. Summarizing the papers reviewed, we observed that the acquisition and analysis were performed based on two approaches: a microcontroller and a portable device (laptop or cell phone). The inertial acquisition rates which were used in over 51 papers reviewed were in the range of 10 Hz [28] to 48 kHz [39]. In the case of lower acquisition frequency, around 10–20 Hz, the proposed studies focused on determining head motion patterns, which has applicability in medical fields [28,40]. The most common acquisition frequency was observed to be within a range of 48 Hz to 100 Hz, with diverse applicability (general HCI solutions, physical and mental activity analyses, sports training, and medical) [23,36,41,42,47,62,64,65,66]. Another range which we determined had been used in the papers reviewed was an acquisition frequency of 120 Hz to 48 kHz. However, this range frequency was rarely used in comparison with 48–100 Hz. Existing studies utilizing this acquisition range are mostly focused on medical fields, such as the study of palsy as a human disorder or to improve the users’ physical and mental state [35,38,56,67].

## 4. Preprocessing and Feature Extraction

In intelligent motion systems, the feature extraction and preprocessing steps consist of some of the main tasks in the process of proposing, designing, and implementing a new solution based on inertial sensors. In a study conducted by Khusainov et al. [68], the authors concluded that the choice of features was more important than other steps because, in the case of low-quality features, the performance of computational models could be directly affected.

Signals from inertial sensors are characterized by noise, which poses difficulties when using these signals in raw form. For this reason, the preprocessing step is critical in the process of developing novel detected head motion systems. In head motion recognition (HMR) systems, the most common approach is characterized by digital and statistical filters [30,33,34,48,50], data normalization [41], and feature extraction [69,70,71]. The feature extraction step explores two domains: time [66] and frequency [43]. Approaches based on handling inertial signals in the time domain are used the most, because these have a small computational time compared with the frequency domain. Approaches based on the identification of features in the time domain describe statistical information using mathematical formulas. In this study, we observed that the most common statistical features were characterized by the calculation of average values, minimum or maximum amplitudes, standard deviations, kurtosis, correlation coefficients, variance, periodicity, and root mean square error (RMSE). In the case of frequency domain, most of the inertial features are based on fast Fourier-transforms (FFTs). Another method is characterized by wavelet transform [34]. This technique is similar to FFT, except that the wavelet transform replaces infinite trigonometric functions with finite wavelet attenuation functions [72]. This method is advantageous because both time and frequency domains are considered. In this review, we found that the most common frequency features are the cross-power spectrum: energy and entropy.

Table 2 presents an overview of the time taken and frequency domain feature extraction applied to the most relevant papers reviewed. For this survey, the computational models were split into two categories: classical machine learning (CMLs) models and deep learning models (DLMs). One key difficulty we met in the review process consisted of information missing (e.g., computational models, preprocessing models, etc.) from the papers selected, affecting the finding of relevant information (e.g., accuracy, subjects, filtering technique, etc.). According to the information presented in Table 2, we observed that the computational models used the most were the classical machine learning models (CML). In the preprocessing and filtering steps, we observed that diverse methods were used for their low complexity as median or average filters, to the more complex methods such as Kalman, Butterworth, Savitzky–Golay, or low/high-pass filters. In most studies related to head motion detection, Kalman-filter-based algorithms were preferred. These techniques use information derived from the expected dynamics of an inertial head motion system to predict a future state given both the current state and a set of control inputs [48,50]. In the selected head motion papers [42,58,59,64,70,73], this estimator was applied to estimate the head orientation in a tridimensional space.

Another aspect is related to the fact that existing studies produced analyses in the time domain. The analyses were performed using data acquired from 1 person [31,32] up to 63 persons [36]. Regarding the age of the volunteers participating in the experiments, there was a wide variation: from 20 years [56] to 68 years old [34]. Most of the values of data acquisition frequency were as follows: 4 Hz [53], 10 Hz [28], 20 Hz [40], 48 Hz [20,36,54,71], 100 Hz [6,36,41,50,55], 125 Hz [38], 128 Hz [35], 200 Hz [47,74], 3.2 kHz [67], and 48 kHz [70]. The maximum sampling rate was found in a paper which proposed an inertial device which enabled psychotherapists to analyze mental-health-related communications [70]. The minimum sampling rate was found in a study which proposed a mouthguard-based inertial safety device for athletes. The classification rate obtained and reported was excellent, suggesting that the inertial sensor was capable of identifying and recognizing head motion patterns. For most papers reviewed, inertial data were acquired from a single inertial sensor placed on the head. Another important aspect is related to the inertial sensor configuration. For this case, we discovered that two main approaches were used. The first approach was characterized by six degrees of freedom sensors, which integrate three axial accelerometer sensors (Acc) and three axial gyroscope sensors (Gyro). The second approach was characterized by nine degrees of freedom sensors, which integrate three axial accelerometer sensors (Acc), three axial gyroscope sensors (Gyro), and three axial magnetometers (Mag). Even though in most of the relevant papers a single inertial sensor was used, head motion patterns were recognized with a good classification rate.

In the reviewed papers, the lowest classification performance was equal to 72.6% [56]; the maximum classification rate was equal to 99.1% [46]. The results were based on data acquired in each independent experiment without using an existing public database. This aspect (missing relevant benchmark datasets) is the main contemporary problem in the field of head motion recognition systems based on inertial sensors. 

**Table 2 jimaging-07-00265-t002:** Preprocessing and feature extraction for HMR systems. “x” means that analyses are applicable to the specified domain. For the case of “-”, this means that is not applicable to that specific domain.

Computational Models	Noise Removal	Time Domain	Frequency Domain	Paper References	Number of Features	Head Recognition Accuracy	Subjects	Number of Sensors	Type of Sensors
CHMR	-	-	x	[34]	1	95%	26	-	Acc and Gyro
CHMR	Median filter	x	-	[75]	-	97.5%	12	1	Acc and Gyro
CHMR	-	x	-	[36]	9	98.56%	63	1	Acc, Gyro, and Mag
DHMR	Butterworth filter	x	-	[57]	7	-	20	1	Acc and Gyro
DHMR	Kalman and low-pass filter	x	-	[58]	7	99.1%	-	1	Acc, Gyro, and Mag
CHMR	Savitzky–Golay and low/high-pass filter	x	-	[46]	4	95%	33	1	Acc and Gyro
CHMR	Kalman filter	x	-	[64]	-	88%	10	2	Acc, Gyro, and Mag
CHMR	-	x	-	[25]	-	92.1%	48	1	Acc, Gyro, and Mag
CHMR	-	x	-	[76]	1	85.66%	6	1	Acc and Gyro
CHMR	-	x	-	[77]	-	78%	5	1	Acc, Gyro, and Mag
CHMR	Average filter	x	-	[28]	-	95.6%	6	1	Mag

In the next section, we present the types of intelligent computational models used in the classification of head motion patterns.

## 5. Computational Motion Models

In the context of proposing and designing precise head motion systems, computational models play an important role, and have the key objective of classifying head motion activity based on data gathered by inertial sensors. In the literature, there are two categories: based on classic machine learning models (CMLs) and deep learning models (DLMs). According to the results reported thus far, we observed that both categories provided a good classification rate based on inertial signals. In this section, each category will be described based on the papers reviewed in this study.

### 5.1. Classical Machine Learning Models

This category of computational models represents a branch of the artificial intelligence research field which has the key purpose of developing algorithms capable of identifying and inferring patterns given an inertial training dataset [78]. The algorithms categorized as such can be divided into two other classes: supervised computational models and unsupervised computational models. The objective of supervised computational models is to design a mathematical model based on the relationship between input and output data to predict future unseen inertial data. On the other hand, in the unsupervised class, the focus is on the identification of head motion patterns from the input dataset without any knowledge of the desired output. Based on the papers related to this study, the most common classical machine learning algorithms are regression models (RMs) [34], random forest (RF) [36], feedforward artificial neural network (FANNs) [58,63,75], dynamic time warping (DTW) [76], decision tree (DTs) [28,36], support vector machines (SVMa) [42,64], k-nearest neighbor (k-NN) [46], fuzzy logic (FL) [79], naïve Bayes classifiers (NBCs) [50,51,62], Euclidian distance classifiers (EDCs) [54], Mahalanobis distance classifiers (MDCs) [54], Gaussian mixture (GM) models [25], Gauss–Newton models (GNMs) [49], adaptive boosting classifiers (ADABs) [80], and multilayer perceptron (MLP) classifiers [81]. Classical machine learning models are usually preferred in the field of head motion recognition or human activity recognition, especially when the training dataset is small, or when a rapid training process is necessary.

In case with a small training database (i.e., data acquired from fewer than 10 people), we observed that the accuracy was in the range of 64.63% [81] to 95.62% [28]. The best classification performances were obtained with KNN (95.62%) [28], SVM (94.80%) [28], DT (92.04%) [28] or RF (91.04%) techniques [80]. In cases with a large training database (data acquired from more than 10 people), we observed that the accuracy was in the range of 89% [23] to 98.61% [36]. For this case, the best classification performances were obtained using RF (98.61%) [36], DT (97.57%) [36], cropped random forest (98.56%) [36], or Gaussian mixture models (92%) [25]. In both situations, we observed that the most suitable computational models for head motion recognition were characterized by bagged computational models. The reported results depend on the complexity of head motion activity and the acquisition period. In the data acquisition period, this process can take from a few seconds or minutes [23,66,77] to the real condition period (15 min [24,60], half an hour or one hour [41,47,70], 3 days [56], or 15 weeks [52]).

Based on this observation, we can conclude that the main advantages of inertial sensors in the process of head motion detection and classification are related to their portability and ease-of-use in real-life scenarios.

### 5.2. Deep Learning Models

Other categories of computational models used in the classification of head motion activity are deep learning models. These have recently become popular in multiple research areas because of their computational performance. The advantage of this category is the fact that this model is based on the idea of data representation, meaning that the desired features can be automatically generated without human intervention. Even though the results reported are excellent, these computational models have a few limitations [82]:They require a large training dataset;They require a large computational period compared to classical machine learning models;The implementation and interpretation of the deep learning models is more difficult than in for classical machine learning models.

Even though the reported results based on deep learning models (DLMs) are excellent [29], in various publications, classical machine learning is preferred, especially when the datasets are small. According to the papers related to the topic of head motion recognition, the most common deep learning models are long short-term memory networks (LSTMs), convolutional neural networks–long short-term memory networks (CNNs–LSTMs), convolutional neural networks (CNNs), bidirectional LSTM networks (BLSTMs) or convolutional neural networks–bidirectional LSTM networks (CNNs–BLSTMs) [29,57], or hidden Markov models [55]. One interesting class of architecture which has contemporary uses in time series processing is CNNs. This model has become a prevalent tool, especially in the field of image processing, from where it has been imported into other research areas. One advantage provided by this model relates to the fact that it imposes local connectivity on the raw data, extracting more important features automatically just by using the training process. The inertial time series in this case is seen as a collection of local signal segments. Each segment is a line of the input image of the CNN. Figure 6 presents an example of such architecture, inputting data information from inertial sensors (accelerometers, gyroscopes, and magnetometers). The core building layer of CNNs is the convolutional layer. Each convolutional layer represents multiple levels of information representations, combined in abstract concepts required to discriminate specific information reflected through inertial signals. Furthermore, a particular function of activation, such as ReLu, can be applied to these convolutional layers. Other layer types, such as activation, pooling, or dropout, are used to reduce the input volume, overfitting, or local invariance. The fully connected layers, presented in traditional feedforward neural networks, are the last CNN layer(s).

In this study, we observed that only 3 papers out of 51 considered determining head motion gestures with deep learning models. Figure 7 shows the distribution of deep learning models and classical machine learning models among the 51 articles reviewed and selected.

In Figure 7, the rest of the papers (indicated by Other) used other computational models or provided improvements to the DLM or CML architecture. According to this observation, we concluded that in cases of head motion recognition, computational models need special adaptation to account for inertial time series. In addition, CML models are usually preferred because of the small datasets. The classification performance reported in the papers reviewed regarding the deep learning models ranged from 80% [29] to 99% [55]. The best deep learning architectures were hidden Markov models (99%) [55], CNNs–BLSTMs (93.62%) [29], CNNs–LSTMs (92.40%) [29], and BLSTMs (91.12%) [29]. On the other hand, the worst results were obtained when using LSTMs (88.18%) [29] and CNNs (80.12%) [29]. The main disadvantage we have noticed in the field of head motion is related to the availability of inertial datasets, which, until now, have not been available to other researchers. This fact represents an impediment in the reproducibility and to improvements of the existing solution. Another point is related to the size of the trainable datasets. In this study, we observed that the number of participants involved in data acquisition varied between 1 person [28,65] and 63 people [36]. Regarding the range of age, there was a wide variation in the age of volunteers who took part in each experiment proposed in the papers reviewed. We observed that the ages ranged from 18 [41] to 68 [34] years old. Another aspect of the papers reviewed is related to how computational models were evaluated (offline vs. online). At present, head motion classifiers can be trained online or offline. An offline training/evaluation (not real-time) is usually seen in the case of applications which are not required to provide rapid feedback to the user or deliver high-performance model classification.

Online evaluations must assist users in a real-time mode to provide fast feedback. Thus, online analyses mean that the classifiers can be trained before being used; however, in the case of offline analyses, the models are built and trained from scratch. Based on the papers selected and reviewed, we discovered that slightly more studies used offline analyses (29/51), with online analyses represented by 24/51 papers. The paper distribution according to the type of analysis is presented in Figure 8.

Based on the distribution presented in Figure 8, we conclude that, in the literature, head motion recognition systems observe the following steps: design, implementation, acquisition, and offline analyses and online evaluations in an equal manner.

## 6. Discussion

In this study, we have presented an overview of head motion recognition methods based on inertial sensors over the last decade. Head motion recognition is a beneficial research area in various fields such as medicine, daily life, or general human–computer interface methods. In addition, the interest shown by researchers for head motion recognition technologies has become popular because new sensor type technologies have been raised and market needs have grown continuously. The main purpose of this field is to improve the quality of life with the utilization of wearable devices incorporating inertial sensors. From the papers reviewed, we observed that the workflow used in specially designed head motion recognition technologies involves four steps. The first step is to determine the topologies of inertial sensors and acquisition methods. The second step is dataset manipulation, including any necessary preprocessing (data collection). The third step includes the identification of computational models and their training. Most studies analyzed used supervised machine learning models based on annotated inertial data (model selection and training). The final step is characterized by computational model evaluation in terms of accuracy, precision, recall, and other metrics. Figure 9 highlights the workflow steps followed in the process of proposing a new system for head motion recognition.

Although in the last decade the number of studies related to head motion recognition has increased significantly, this research field still has multiple aspects which need to be explored. One important impediment we observed during the literature review is related to the reproducibility of results. Most existing studies do not publish their datasets; therefore, this aspect is a hinderance for the wider research community during the identification of the best computational methods or benchmarking the results. Another consequence of the lack of public head motion datasets is the low capability of head motion generalization because of the data collected in a controlled environment. This aspect is due to a small number of test activities or small amount of data acquired from the volunteers involved. Among the 51 papers reviewed, all datasets were created from scratch, acquiring inertial data from a minimum of 1 person [37,55], up to 63 people [36]. Most papers analyzed used healthy volunteers to perform the desired analyses. Although the number of studies testing the solution proposed in real conditions is pretty low, in the solutions focused on the medical field, we observed that several real patients had been involved in experiments. These applications monitored medical problems such as head tremor [34], cerebral palsy [54], and fall detection [40]. The wearable head motion devices proposed acquired inertial signals with frequency rates ranging from 4 Hz [53] to 48 kHz [70]. For detecting head motion during various daily activities, wearable devices usually work with an acquisition frequency between 48 Hz [20,36,54,71] and 100 Hz [6,36,41,50,55]. This aspect suggests that head motion patterns can be detected by each wearable device based on inertial sensors available on the market.

Another important aspect is related to sensor placement. Consequently, the most common approach was when the inertial sensor was placed on the left or right of the forehead (20.73%), on the forehead (18.86%), and on top of the head (16.98%). Other areas commonly used are the back of the head (11.32%), ear (11.32%), eye or neck (7.52%), and other head areas (5.64%). The most common types of sensors used are 9DOF inertial sensors (accelerometer, gyroscope, and magnetometer), 6DOF inertial sensors (accelerometer and gyroscope), and 3DOF accelerometer inertial sensors. During the technical analysis, we observed that the topologies of inertial sensors and their placement on different areas of the head can affect the classification performance of the computational models. Regarding head motion recognition models, the results demonstrated that classical machine learning models (CMLs) are used more widely than deep learning models (DLMs). This distribution is presented in Figure 7. CMLs are the most common approach among head motion analyses, because they require a small amount of training data, as well as lower computational requirements. Another advantage of this category is related to the complexity of head motion activity, which is low in comparison with the requirements for DLMs. In DLMs, this architecture enables the recognition of more complex activities, and it does not require an additional preprocessing step. In CML models, the architectures used most in the field of head motion detection are regression models (RMs) [34], random forest (RF) [36], feedforward artificial neural networks (FANNs) [58,63,75], dynamic time warping (DTW) [76], decision tree (DTs) [28,36], support vector machines (SVMs) [42,64], k-nearest neighbor (k-NN) [46], fuzzy logic (FL) [79], naïve Bayes classifier (NBC) [50,51,62], Euclidian distance classifiers (EDCs) [54], Mahalanobis distance classifiers (MDCs) [54], Gaussian mixture models (GMs) [25], Gauss–Newton models (GNMs) [49], adaptive boosting classifiers (ADABs) [80], and multilayer perceptron (MLP) classifiers [81]. As for DLM models, the most common deep learning models are long short-term memory networks (LSTMs), convolutional neural networks–long short-term memory networks (CNNs–LSTMs), convolutional neural networks (CNNs), bidirectional LSTM networks (BLSTMs), convolutional neural networks–bidirectional LSTM networks (CNNs–BLSTMs) [29,57], and hidden Markov models [55]. In terms of classification performance, we observed that in the case of CML models, the best models incorporated KNN (95.62%) [28] and SVMs (94.80%) [28] for small datasets (i.e., data acquired from fewer than 10 volunteers). In the case of a large dataset (i.e., data acquired from more than 10 volunteers), we obtained the best classification performances using RF (98.61%) [36] and DT (97.57%) techniques [36]. Based on the papers selected for this study, we observed that the process of computational model selection (DLM or CML) is generally based on the computational requirements and on the size of the available training (labeled) dataset. In terms of head motion activity, we observed that the minimum number of recognized head activities was 3 [28]; the maximum number of head activities studied in one specific study was 20 [62]. Consequently, the solutions studying or proposing wearable devices are characterized by a lack of standardization related to a heterogeneous set of head activities performed by users with different head characteristics.

## 7. Conclusions and Research Direction

### 7.1. Conclusions

This paper has presented a literature overview for the last decade of state-of-the-art head motion monitoring systems based on inertial sensors. This study focused on determining the acquisitional methods used, structure of the prototypes, preprocessing steps, computational methods, and the techniques used to validate these systems. Regarding our main scope analyzed in this review (head motion recognition systems), we can conclude that the actual HMR solutions have multiple disadvantages compared to other techniques (e.g., vision-based). The main disadvantage of the current studies is related to the limited availability of datasets, which leads to a low reproducibility of existing solutions or results. Another disadvantage is related to the limited generalization of solutions, because most of the proposed solutions were performed in controlled experimental conditions (i.e., in a laboratory and involving healthy volunteers). The third point which we observed based on the papers reviewed is represented by the wearability of the devices proposed, which, in several cases, are difficult to use in real conditions. Head motion recognition can be a beneficial research area in various fields, such as medicine, daily life, or human–computer interface methods. In the medical field, the detection of head motion patterns is important in the diagnosis of various diseases or to support ill or elderly persons. Such examples which could be considered are the detection of head tremors, detecting involuntary falls, vestibular rehabilitation, physical and mental analyses, or human balance and orientation. In addition, the head motion patterns could be essential for assistance systems. Such systems could help people interact independently with a mechanical system, such as paralyzed persons operating a wheelchair. For applications in daily life, the detection of head motion is important in fields such as sports training, sleep quality, or monitoring drivers’ attention.

During our study, we observed that head motion systems based on inertial sensors could be beneficial in the field of augmented reality. We expect considerable developments in this field in the near future, facilitated by the COVID-19 pandemic and the interest of private companies to develop a social media metaverse world. Thus, head motion detection based on inertial sensors could be considered a niche opportunity for multidisciplinary research. Based on the papers analyzed, we identified four trends in head motion analysis, summarized below.

The first trend in the literature is characterized by studies focused on determining the topologies and acquisition methods of inertial sensors. The second trend is characterized by studies focused on data engineering, including any necessary preprocessing method (data collection). The third trend includes studies focused on the identification and proposal of new computational models for analyzing head motion. The fourth trend is characterized by studies focused on evaluating existing computational models in terms of head motion activity or, more generally, in the recognition of human activity. Regarding the topology of inertial sensors, we observed that the most common approaches are based on six degrees of freedom (6DOF) and nine degrees of freedom (9DOF). In most of the papers reviewed, a single inertial sensor was included in the final prototype. Among the 51 papers reviewed, all datasets were created from scratch, acquiring inertial data from a minimum of 1 person up to 63 people. Most experiments used healthy volunteers to perform the desired analyses under controlled conditions (i.e., in a laboratory). Even though the studies proposed are promising, no datasets have been made public for the wider research community. Therefore, we consider that major future efforts must focus on improving collaboration and cooperation among researchers to make their work public to the global community.

### 7.2. Research Direction

Based on the papers reviewed, we have determined a few potential research directions in the field of head motion pattern analyses using inertial sensors. One possible future research direction is to propose and analyze various generalization methods for computational models. Thus, it will be possible to generalize a heterogeneous set of head motion activities performed by a diverse set of users, avoiding the specificity of each user in the development of new head motion systems. A possible solution for this research direction could be in reusing the knowledge acquired in a specific field (e.g., medical field, general HCI, daily activity, etc.) to solve a similar problem. One example could be that the information acquired from the head can be reused for understanding other body motions with the help of inertial sensors, or even based on other sensor topologies. Another research direction could be the field of sensor fusion methods. In this direction, the information from inertial sensors could be a fusion with information from other wearable sensors (e.g., GPS sensors, EMG sensors, etc.). Based on this approach, the reliability and accuracy performance issues of the solutions proposed could be analyzed in the field of head motion recognition, or others. This method can be beneficial for the detection of suitable sensor topology in determining specific head motion patterns.

## Figures and Tables

**Figure 1 jimaging-07-00265-f001:**
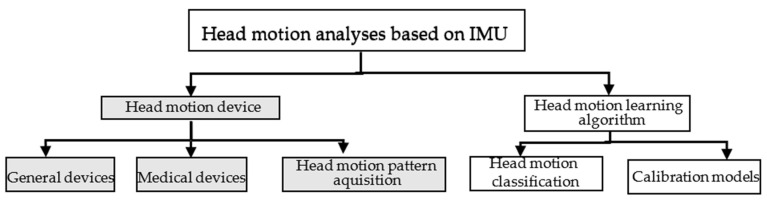
Taxonomy of head motion recognition applications.

**Figure 2 jimaging-07-00265-f002:**
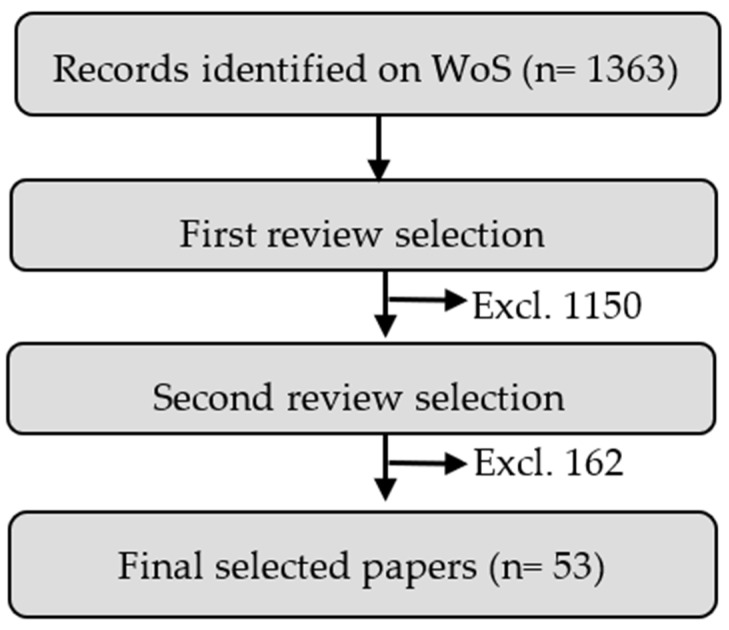
Paper selection steps.

**Figure 3 jimaging-07-00265-f003:**
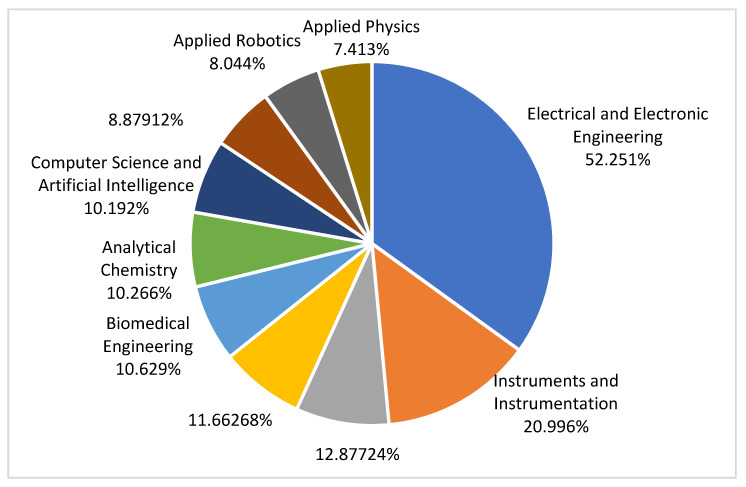
Distribution of studies by technical area.

**Figure 4 jimaging-07-00265-f004:**
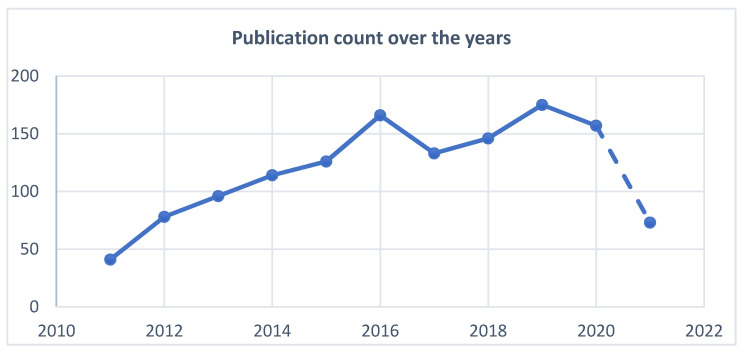
Publication count over the years.

**Figure 5 jimaging-07-00265-f005:**
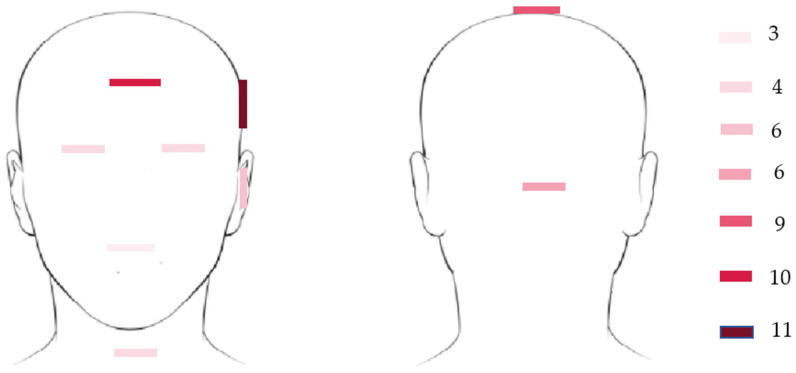
Distribution of inertial sensors on the head.

**Figure 6 jimaging-07-00265-f006:**
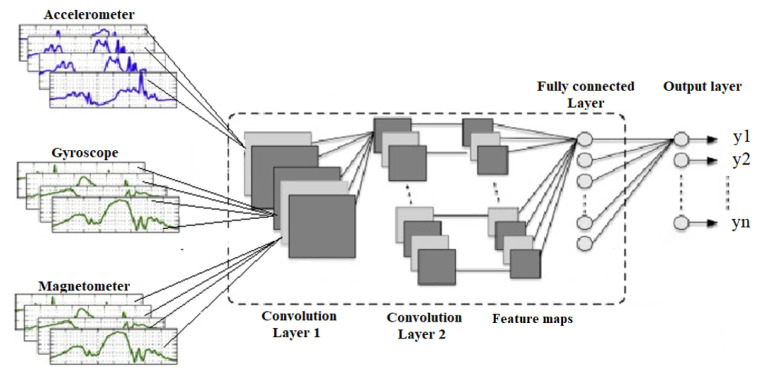
Example of a CNN computational model for inertial signal classification.

**Figure 7 jimaging-07-00265-f007:**
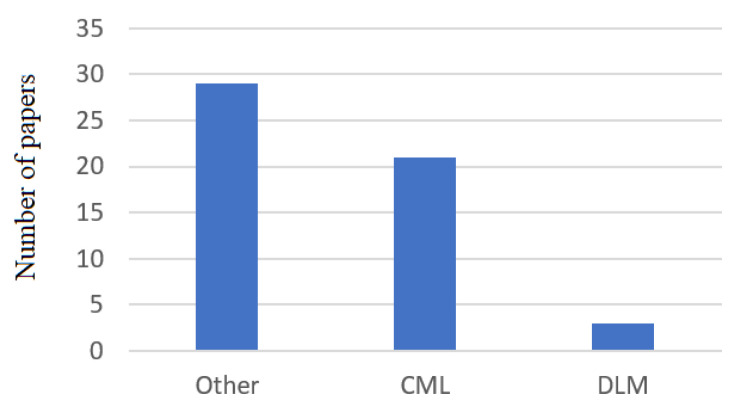
Distribution of computational models used in head motion recognition.

**Figure 8 jimaging-07-00265-f008:**
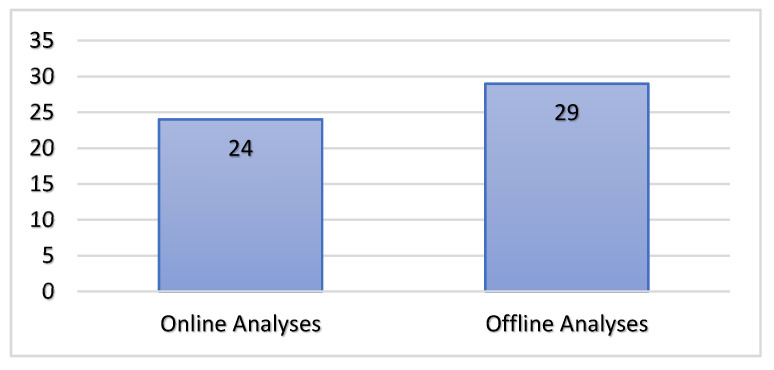
Distribution of studies related to online vs. offline analyses.

**Figure 9 jimaging-07-00265-f009:**
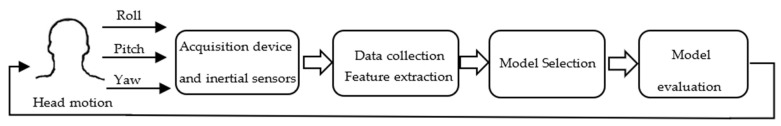
Workflow for implementing head motion recognition solutions based on inertial sensors.

**Table 1 jimaging-07-00265-t001:** Existing human activity recognition surveys.

Paper Reference	Publication Year	Main Focus	Body Part	Reviewed Papers
[5]	2020	Activity recognition methods	Full body	8
[6]	2020	Classification of the position and number of inertial sensors	Full body	58
[7]	2019	Deep learning Human activity recognition (HAR)	Full body	75
[8]	2019	HAR in healthcare	Full body	256
[9]	2019	HAR in a multi-data system	Full body	309
[10]	2018	Smartphone-based HAR	Full body	273
[11]	2018	Classification algorithms for HAR systems	Full body	-
[12]	2017	Smartphone-based HAR	Full body	37
[13]	2016	Wearable HAR	Full body	225
[14]	2013	Wearable HAR	Full body	28
[15]	2020	Classification algorithms for HAR systems	Full body	147
[16]	2016	Activity recognition methods	Full body	36
[17]	2020	Activity recognition methods	Full body	95

## Data Availability

Not applicable.

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
