# Peer review of "Using Inertial Sensors to Determine Head Motion—A Review"

_2313-433X, 2021, doi:10.3390/jimaging7120265_

Round 1

Reviewer 1 Report

The authors have carried out a meta-analysis of studies on motion analysis using head-mounted IMUs. Since consumuer products includes IMU sensors such as AirPods or JINS MEME etc., which can be worn on the head by ordinary people on a daily basis, are just now spreading in the market, the result will be very useful to study their application.

Minor points

  • The word "warble" in the 39th line, is "wearable"?
  • In the Fig. 4, it would be better to use a dotted or lighter line for 2021, because the publication count is not decreasing significantly.
  • What does "CML" stand for? In the lines 420th and 426th, you wrote "classic head motion models", but elsewhere wrote "classic machine learning models".
  • The word "DML" is "DLM" correct in the lines 421th and 458th, isn't it?
  • In the lines 443rd and 621st, "50KHz" (or "50kHz") is correct.

Suggestion

The research methods and analysis are generally considered reasonable.
However, it may be better if new keywords or search sources are considered.
I guess it wasn't cited because it wasn't a full paper, but I felt that there should be more citations of key words for new concepts such as "earable computing" or "eyewear (+ motion)".

Author Response

Dear reviewer,

Below you have the comments according you're review points:

  • The word "warble" in the 39th line, is "wearable"?
    • Answer:  You have right, was a wrong spelling. The correction has been implemented.
  • In the Fig. 4, it would be better to use a dotted or lighter line for 2021, because the publication count is not decreasing significantly.
    • Answer: The suggestion was accepted and implemented in the paper. 
  • What does "CML" stand for? In the lines 420th and 426th, you wrote "classic head motion models", but elsewhere wrote "classic machine learning models".
    • Answer: Thank you for your suggestion. Is a wrong spelling in a text, the form from lines 420th and 426th will be replaced with the format of CML(Classical Machine Learning models)
  • The word "DML" is "DLM" correct in the lines 421th and 458th, isn't it?
    • Answer: Thank you for your suggestion. Yes, was a wrong spelling. The correction has been implemented.
  • In the lines 443rd and 621st, "50KHz" (or "50kHz") is correct.
    • Answer: Thank you for your suggestion. Both could be valid, but for uniformity of the information presented in the paper I replaced in text with kHz instead of KHz.
  • However, it may be better if new keywords or search sources are considered. I guess it wasn't cited because it wasn't a full paper, but I felt that there should be more citations of key words for new concepts such as "earable computing" or "eyewear (+ motion)".
    • Answer: Thank you for your suggestion. New keywords have been attached to the paper.

For more details see the attached files.

Reviewer 2 Report

This paper give a review of the head activity recognition, but the paper is not well organized. It’s not suitable to publish for a journal paper.

  1. In line 34-35, “They have multiple contributions 34 such as reading of air chemical components to determine the quality of air, translate the 35 degree of heat into temperature, use mechanical”, these sentences has fewer related to your works. It’s better for the author to delete the paragraph with no scene.
  2. Section 1, introduction, only one paragraph, it’s better for the author re-organize them.
  3. As to the related work, it’s better for the author to provide a table, in order to better compare the literature researches.
  4. The discussion and future research directions are very important for a review paper. Obviously, the author didn’t give sufficient description about these sections.

Author Response

Dear reviewer,

Below you have the comments according you're review points:

  • In line 34-35, “They have multiple contributions 34 such as reading of air chemical components to determine the quality of air, translate the 35 degree of heat into temperature, use mechanical”, these sentences has fewer related to your works. It’s better for the author to delete the paragraph with no scene.
    • Answer: Thank you for your suggestion. The observation has been accepted and these sentences were deleted from the paper.
  • Section 1, introduction, only one paragraph, it’s better for the author re-organize them.
    • Answer: Thank you for your suggestion. The observation has been accepted; the chapter has been formatted including more than one paragraph.
  • As to the related work, it’s better for the author to provide a table, in order to better compare the literature researches.
    • Answer: Thank you for your suggestion. See Table 1 that was added in the introduction section.
  • The discussion and future research directions are very important for a review paper. Obviously, the author didn’t give sufficient description about these sections.
    • Answer: Thank you for your suggestion. We added more information in the section “Conclusion” that from our perspective is the next part of the chapter “discussion”.

Reviewer 3 Report

This article presents a review of head motion detection (a subdomain of human activity recognition) focusing on using inertial sensors. State-of-the-art literatures from 4 databases (IEEE Xplore, Elseview, MDPI an Springer) were reviews with well organization of academic writing.

  1. The authors should explain better the novel contribution of the article in the introduction.
  2. In the introduction, the authors did not explain the usefulness of head motion detection. To improve the readability, the authors should add the usefulness including practical applications.
  3. Two diagrams in Figure 3 are redundant that explain in the same thing, the authors should combine them in only one diagram.
  4. In the Section 3, the authors should review the process of data collection for head motion detection, also some benchmark datasets for head motion detection should be explain in this section. A summary table of datasets should be shown with information such as number of subjects, number of sensors, types of sensors, sampling rate, etc.
  5. The authors stated about the CNN that is an interesting model in time series processing (line 524-525). In my opinion, there are other DL models (such as LSTM, GRU) that outperforms the CNN. Therefore, the authors should provide reasons for this statement.
  6. The Figure 6 should give more explanation about CNN’s computation.
  7. In the Section 7, everything was explained in one paragraph that make difficult to read. The authors should separate into two subparagraph: 1) conclusion and 2) future research direction.

Author Response

Dear reviewer,

Below you have the comments according you're review points:

1. The authors should explain better the novel contribution of the article in the introduction.

     Answer:  Thank you for your suggestion. A new phrase was added into introduction chapters, see following phrase: “The novelty provided by our proposal is represented by the fact that our study is focusing on the literature inspection related to the existent methodologies applied on an understanding motion from a specific body part (head motion). Until now, we didn't discover a similar approach in the literature, even for other parts of the human body. Most existing surveys are focused in the following: activity recognition methods, classification algorithms for activity recognition systems or wearable inertial systems”. See information starting with 65-73

2. In the introduction, the authors did not explain the usefulness of head motion detection. To improve the readability, the authors should add the usefulness including practical applications.

Answer: Thank you for your suggestion. The more detailed information related to the importance of understanding of head motion have been introduced starting with line 12-15.

3.Two diagrams in Figure 3 are redundant that explain in the same thing, the authors should combine them in only one diagram.

Answer: Thank you for your suggestion. The second picture was deleted from the paper.

4. In the Section 3, the authors should review the process of data collection for head motion detection, also some benchmark datasets for head motion detection should be explain in this section. A summary table of datasets should be shown with information such as number of subjects, number of sensors, types of sensors, sampling rate, etc.

Answer: Thank you for your suggestions and comments. From the beginning of our research, in the field of using inertial sensors to determine the head's position and movement, we have encountered this big problem: the lack of benchmark databases with such signals that we can use. This fact led us to build our database based on the systems developed by us.

Regarding your specific suggestion, we addressed it by adding additional information in Table 2 (number of sensors and types of sensors). Moreover, additional information is presented in lines 468 – 490.

5.The authors stated about the CNN that is an interesting model in time series processing (line 524-525). In my opinion, there are other DL models (such as LSTM, GRU) that outperforms the CNN. Therefore, the authors should provide reasons for this statement.

 Answer: Thanks for the feedback. You are right; there are many other DL neural networks with higher performances. We refined our presentation to point more explicitly the CNN class of architectures – see the lines 572  – 578.

6. The Figure 6 should give more explanation about CNN’s computation.

Answer: We thank the reviewer for the opportunity to explain the CNN concepts better. As a direct result, we amended the version of the manuscript adding the lines 578 (“The core building layer of CNN is the convolutional”) – 587.

7. In the Section 7, everything was explained in one paragraph that make difficult to read. The authors should separate into two subparagraph: 1) conclusion and 2) future research direction.

Answer: Thanks for the feedback. The suggestion was accepted and implemented in the final version of the paper.

Round 2

Reviewer 2 Report

The authors already revise the paper according to the reviewers comments.